

# Dental caries as a risk factor for bacterial blood stream infection (BSI) in children undergoing hematopoietic cell transplantation (HCT)

Dawud Abduweli Uyghurturk[1,2,*], Ying Lu[3,*], Janelle Urata[1,2], Christopher C. Dvorak[3] and Pamela Den Besten[1,2]

[1] Orofacial Science, University of California, San Francisco, San Francisco, CA, United States
[2] Center for Children's Oral Health Research, School of Dentistry, University of California, San Francisco, San Francisco, California, United States
[3] Division of Pediatric Allergy, Immunology and Bone Marrow Transplant, University of California, San Francisco, San Francisco, California, United States
* These authors contributed equally to this work.

Corresponding author
Pamela Den Besten,
pamela.denbesten@ucsf.edu

## ABSTRACT

**Background:** Hematopoietic cell transplantation (HCT) is a potentially curative therapy for a wide range of pediatric malignant and nonmalignant diseases. However, complications, including blood stream infection (BSI) remain a major cause of morbidity and mortality. While certain bacteria that are abundant in the oral microbiome, such as *S. mitis*, can cause BSI, the role of the oral microbial community in the etiology of BSI is not well understood. The finding that the use of xylitol wipes, which specifically targets the cariogenic bacteria *S. mutans* is associated with reduced BSI in pediatric patients, lead us to investigate dental caries as a risk factor for BSI.

**Methods:** A total of 41 pediatric patients admitted for allogenic or autologous HCT, age 8 months to 25 years, were enrolled. Subjects with high dental caries risk were identified as those who had dental restorations completed within 2 months of admission for transplant, or who had untreated decay. Fisher's exact test was used to determine if there was a significant association between caries risk and BSI. Dental plaque and saliva were collected on a cotton swab from a subset of four high caries risk (HCR) and four low caries risk (LCR) children following pretransplant conditioning. 16SrRNA sequencing was used to compare the microbiome of HCR and LCR subjects and to identify microbes that were significantly different between the two groups.

**Results:** There was a statistically significant association between caries risk and BSI ($p < 0.035$) (Fisher's exact test). Multivariate logistic regression analysis showed children in the high dental caries risk group were 21 times more likely to have BSI, with no significant effect of age or mucositis severity. HCR subjects showed significantly reduced microbial alpha diversity as compared to LCR subjects. LEfse metagenomic analyses, showed the oral microbiome in HCR children enriched in order Lactobacillales. This order includes *Streptococcus* and *Lactobacillus*, both which contain bacteria primarily associated with dental caries.

**Discussion:** These findings support the possibility that the cariogenic microbiome can enhance the risk of BSI in pediatric populations. Future metagenomic analyses to measure microbial differences at, before, and after conditioning related to caries risk,

may further unravel the complex relationship between the oral microbiome, and whether it affects health outcomes such as BSI.

# INTRODUCTION

Hematopoietic cell transplantation (HCT) is a potentially curative therapy for a wide range of pediatric malignant and nonmalignant diseases. However, complications associated with HCT conditioning regimens and post-HCT immunocompromised status, such as mucositis and blood stream infection (BSI), remain a major cause of morbidity and mortality (*Dahllof, 2008*; *McNamara et al., 2018*). The incidence of BSI is reported to vary from 22% to 55.8% and is related to prolonged neutropenia, mucosal damage, and extensive use of venous central lines (*Gustinetti & Mikulska, 2016*).

In this study we tested the hypothesis that dental caries contributes to BSI risk in pediatric HCT recipients. This possibility is supported by the study by *Badia et al. (2020)*, which showed daily use of xylitol wipes in children undergoing HCT significantly reduced dental plaque, mucositis and bacteria known to cause BSI. Xylitol reduces dental plaque by substituting for carbohydrates consumed by the acid and extracellular polysaccharide producing cariogenic bacteria, *S. mutans* (*Söderling & Pienihäkkinen, 2020*). Reduced extracellular polysaccharides means that plaque is less adherent and more easily removed from the tooth surface (*Söderling & Pienihäkkinen, 2020*). This suggests that changes in cariogenic bacteria contributes to BSI risk.

To address a possible role of a cariogenic microbiome as a risk factor for BSI, we enrolled 41 pediatric allogenic or autologous HCT recipients, age 8 months to 25 years, and classified them as either high caries risk (HCR) or low caries risk (LCR) based on pretransplant dental clearance records. We used 16SrRNA sequencing to compare the plaque/saliva microbial composition in a subgroup of 4 HCR and 4 LCR subjects.

# MATERIALS AND METHODS

## Subject enrollment and sample collection

Study patients: The University of California San Francisco granted approval to carry out this study within its facilities under IRB protocol # 15-18297. Forty-one of 43 pediatric HCT recipients, age 8 months to 25 years, treated in the University of California, San Francisco (UCSF) Benioff Children's Hospital and approached for participation in the study, were enrolled between the years 2016 and 2019. Inclusion criteria were HCT recipients who could cooperate with study procedures, and parents and patients willing to participate and sign informed consent and assent forms. Exclusion criteria were patients or parents unable to understand or participate in study procedures, or allergies to multiple hygiene and cosmetic products. Written consents by parents and assent by children were obtained for all subjects.

Dental caries risk was determined by retrospective examination of dental clearance records. Past caries experience is a robust predictor of the disease (*Philip, Suneja & Walsh, 2018*), and we classified participants with dental clearances stating "clear/no caries detected/minimal decay", or extractions of third molars with no other mention of dental caries or restorations, as low caries risk (LCR). Subjects with dental clearances stating "treatment for multiple dental caries" or indications of untreated decay 2 months prior to admission for transplant, were classified as high caries risk (HCR).

All patients were administered prophylactic levofloxacin from Day-2 until neutrophil recovery (*Alexander et al., 2018*). Mucositis was assessed from the medical records with daily assessments by nurses and physicians recorded according to the WHO classification standards. Bacterial blood stream infections (BSI) were defined as any positive bacterial culture of (central line) blood samples during the transplantation hospital stay.

Oral samples were collected by using a cotton swab to collect saliva and plaque, by rubbing the cotton swab across the tooth surfaces and under the tongue. All samples were stored at −80 °C until they were submitted for 16S rRNA sequencing.

## 16S rRNA sequencing and bioinformatics

Oral samples were taken from the entire cohort and sent to uBiome for sequencing. Thirty-one fastq files were sent back to us from uBiome.

Samples were processed and sequenced by uBiome (San Francisco, CA, USA) as reported by *Almonacid et al. (2017)*. Briefly, the V4 variable region of the 16S rRNA genes was amplified by PCR using universal primers (515F: GTGCCAGCMGCCGCGGTAA and 806R: GGACTACHVGGGTWTCTAAT). The primers contained Illumina tags and barcodes. PCR products were pooled, column-purified, and size-selected through microfluidic DNA fractionation. Sequencing was performed in a pair-end modality on the Illumina NextSeq 500 platform rendering $2 \times 150$ bp pair-end sequences. The sequences obtained from uBiome included primers within the 150 bp sequence, and when the primers were removed, the remaining sequences were up to 125 bp in length. With this length of sequence, there were not overlapping sequences greater than 10 base pair, which is not ideal and therefore lead us to use only forward reads for our analysis.

Bioinformatic analyses were done using QIIME2 (version 2021.4) and R programming language (version 4.1.2), on forward reads of the sequencing data. Reads containing more than two expected errors and more than eight consecutive same nucleotide repeats were discarded, and the raw sequences were then truncated to 125 pb, denoised, chimera filtered, clustered into sequence variants. The taxonomy was assigned to operational taxonomic units (OTUs; 97% clustering) using the HOMD (Human Oral Microbiome Database version15.2) database with QIIME2 pipeline default settings. After analyzing rarefaction curves and the sampling depth that would retain the most feature in QIIME pipeline we have left eight samples for analyses. Feature table, rooted tree and taxonomy files were then imported into R for diversity, PCoA and differential expression analysis. Phyloseq package (version 1.30.0) was used for diversity and PCOA analyses. We further evaluated the differentially expressed taxa with "The Linear Discriminative Analysis (LDA) Effect Size (LEfSe) program of the Galaxy environment".

**Table 1 Demographics of study population (_n_ = 41).**

|  | Low caries risk (_n_, %) | High caries risk (_n_, %) | _p_-value |
|---|---|---|---|
| Subject number | 27 | 14 |  |
| Gender (female) | 8 (29.6) | 9 (64.3) | 0.072 |
| Age (year, mean (range)) | 9.6 (0.7–25) | 12.6 (1.6–21) | 0.239 |
| Diagnosis |  |  | 0.439 |
| Leukemia | 7 (25.9) | 7 (50.0) |  |
| Lymphoma | 2 (7.4) | 1 (7.1) |  |
| Non-malignant condition* | 4 (14.8) | 2 (14.3) |  |
| Solid Tumor | 14 (51.9) | 4 (28.6) |  |
| HCT type |  |  | 0.271 |
| Allogenic | 11 (40.7) | 9 (64.3) |  |
| Autologous | 16 (59.3) | 5 (35.7) |  |
| Stem cell source |  |  | 0.835 |
| Bone marrow | 4 (14.8) | 1 (7.1) |  |
| PBSC | 23 (85.2) | 13 (92.9) |  |
| Mucositis grade 0–4** (mean (SD)) | 2.1 (1.4) | 2.4 (1.2) | 0.424 |

Notes:
* Non-malignant condition: Severe aplastic anemia, Severe congenital neutropenia, and Primary immunodeficiency disorder.
** WHO grade •Grade 0 (none) – No oral mucositis; •Grade 1 (mild) – Erythema and soreness; •Grade 2 (moderate) – Ulcers; able to eat solid food; •Grade 3 (severe) – Ulcers; but requires liquid diet (due to mucositis); •Grade 4 (life-threatening) – Ulcers; alimentation not possible due to mucositis.

## Statistical analysis

Analysis of BSI as the dependent or the outcome variable of interest, was constructed as a 'yes/no', and was compared in the two groups using a Fisher's Exact test. A correlation matrix was done comparing BSI, caries risk, age, gender, HCT-type (autologous or allogenic), cell type (peripheral or bone marrow), length of time for mucositis, and highest mucositis grade. A multiple regression analysis was done to further investigate the effect of age and mucositis, on caries risk and BSI. The specific logistic regression model fitted to the data was: Logit (BSI) = $b_0 + b_1X_1 + b_2X_2 + b_3X_3$. All data analyses were conducted using R version 4.1.2. Effect Size (LEfSe) was performed using the Galaxy web application using default settings. Statistical significance was defined as $p \leq 0.05$.

## RESULTS

### Demographics

In our study group, 27 of 41 subjects (65.9%) were classified as LCR and 34.1% were HCR. There were no significant differences ($p < 0.05$) between the groups in relative to age, gender, diagnosis, stem cell source, or maximum mucositis grade (Table 1).

### Incidence of BSI

All BSI incidences were between post-HCT and prior to ANC (absolute neutrophil count) recovery/engraftment. Among them 57% were allogeneic HCT recipients (see Table 2). High caries risk was significantly associated with subsequent BSI during transplantation

**Table 2  Demographics of BSI patients (*n* = 6).**

|  | Low caries risk (*n*, %) | High caries risk (*n*, %) | *p*-value |
|---|---|---|---|
| Subject number | 2 | 4 | |
| Gender (female) | 1 (50.0) | 2 (50.0) | 1.000 |
| Age (year, mean (range)) | 1.6 (1.3–2) | 8.2 (1.6–19) | 0.308 |
| Diagnosis | | | 0.223 |
| Leukemia | 0 (0.0) | 2 (50.0) | |
| Non-malignant condition | 0 (0.0) | 1 (25.0) | |
| Solid Tumor | 2 (100.0) | 1 (25.0) | |
| HCT type | | | 0.386 |
| Allogenic | 0 (0.0) | 3 (75.0) | |
| Autologous | 2 (100.0) | 1 (25.0) | |
| Stem cell source | | | 1.000 |
| BM | 0 (0.0) | 1 (25.0) | |
| PBSC | 2 (100.0) | 3 (75.0) | |
| Mucositis grade | | | 0.223 |
| Grade 0 | 1 (50.0) | 0 (0.0) | |
| Grade 1–2 | 0 (0.0) | 2 (50.0) | |
| Grade 3 | 1 (50.0) | 2 (50.0) | |

($p = 0.035$) (Fig. 1). Correlation analysis of other possible risk factors for BSI, showed only BSI to be significantly positively correlated with caries risk, ($p < 0.05$) (Fig. 2). Multiple regression analysis of the association of BSI with age, caries risk, and mucositis grade, showed an even more significant association between BSI and caries risk ($p = 0.02$) (Table 3).

## Microbiome analysis

We compared the oral microbiome of a combination of saliva and plaque collected from four HCR and four LCR subjects following pretransplant conditioning (see Table 4). 16S rRNA sequencing yielded mean number of 88,610.875 reads per sample (range 40,520.0–187,555.0) after quality control processes. Raw sequencing files were deposited in GEO with submission number: GSE199991. Clustering was carried out at 97% identity to yield 478 distinct OTUs with a length of 125 bp. The HOMD framework allowed 98% of the OTUs to be assigned to uniquely identifiable genus-level groups. 454 OTUs classified as bacteria remained after filtering, and were assigned as Eukaryotes, Archaea, chloroplasts, and mitochondria. The most abundant phylum, family, and genus, across all samples at >1% abundance, are shown in Fig. 3, and summarized in Table 5, which shows the most abundant top four taxa across all samples.

Following conditioning, the alpha diversity in the LCR group was significantly greater than in the HCR low caries risk group ($p < 0.05$) (Fig. 4A). Beta diversity metrics showed no clear clustering by community composition by principal coordinate analysis (Figs. 4B–4D).

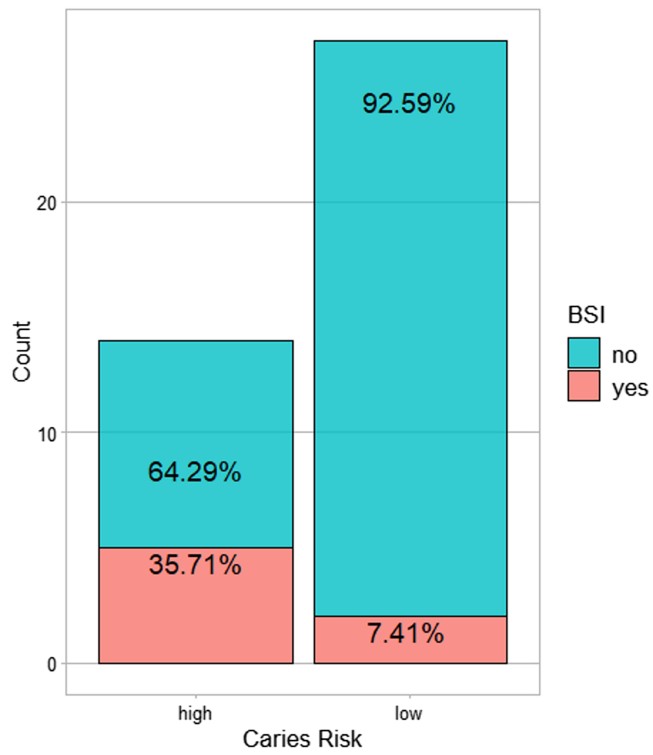

**Figure 1 The number of subjects with BSI in the high caries risk as compared to low caries risk.**

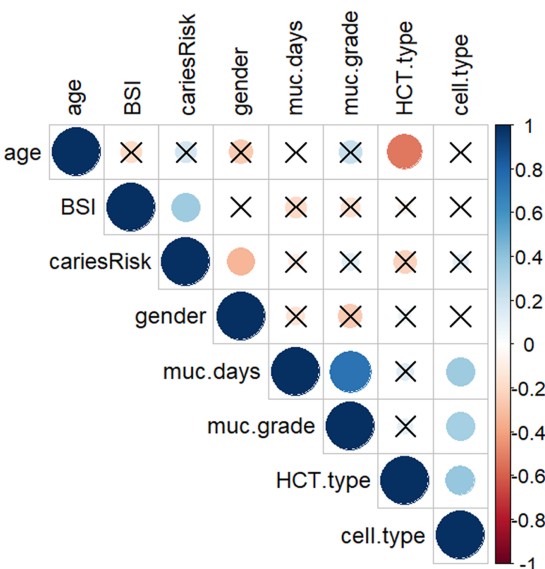

**Figure 2 Correlation matrix plot showing the correlation between each pair of variables.** A dot-re-presentation was used where blue represents positive correlation and red negative. The larger the dot the larger the correlation. BSI showed significant correlation with high caries risk. High caries risk was coded as 1, low caries risk as 0; Gender: male coded as 1, female coded as 0; HSCT-type: hematopoietic stem cell transplant type, auto type coded as 1, allo type coded as 0; Cell-type: PBSC coded as 1, BM coded as 0. Non-significant ($p > 0.05$) associations are crossed out.

**Table 3 Association with BSI.**

| Predictors | Estimate | Odds ratios | 95% confidence interval | *p*-value |
|---|---|---|---|---|
| | **BSI** | | | |
| (Intercept) | −0.959 | 0.38 | [0.05–2.09] | 0.301 |
| Age | −0.130 | 0.88 | [0.72–1.02] | 0.124 |
| Caries risk | 3.063 | 21.39 | [2.44–427.55] | 0.016 |
| Mucositis grade | −0.487 | 0.61 | [0.25–1.32] | 0.225 |

Notes:
Observations 41.
$R^2$ Tjur 0.27.

**Table 4 Demographics of subjects for microbiome analyses.**

| | Low caries risk (*n*, %) | High caries risk (*n*, %) | *p*-value |
|---|---|---|---|
| Subject number | 4 | 4 | |
| Gender (female) | 1 (25.0) | 3 (75.0) | 0.48 |
| Age (year, mean (range)) | 16.0 (12–21) | 13.1 (5.4–21) | 0.5 |
| Diagnosis | | | 0.41 |
| Leukemia | 2 (50.0) | 4 (100.0) | |
| Lymphoma | 2 (50.0) | 0 (0.0) | |
| HCT type | | | 1.0 |
| Allogenic | 3 (75.0) | 4 (100.0) | |
| Autologous | 1 (25.0) | 0 (0.0) | |
| Stem cell source | | | |
| PBSC | 4 (100) | 4 (100) | |
| Mucositis grade | | | 0.48 |
| Grade 1–2 | 1 (25.0) | 3 (75.0) | |
| Grade 3–4 | 3 (75.0) | 1 (25.0) | |

We next performed LEfSe and DESeq2 analyses to identify individual microbial taxa that differed between groups. Both LEfSe and DESeq2 were performed from phylum to species level (Fig. S2). For the comparison HCR vs LCR, in DESeq2 analysis, 22 taxa were enriched in the LCR group, while eight taxa were enriched in the HCR group (Fig. S2). LEfSe showed 56 taxa enriched in the LCR group while two taxa were enriched in HCR group (Fig. 5, Fig. S3.4). Fifteen taxa were enriched in the LCR group, order Lactobacillales and class Bacilli were the taxa significantly enriched in HCR caries group in both analysis (Table 6).

While differences were only significant at the order and class level, relative abundances of Streptococcus and Lactobacillus at the genus level showed relatively high levels of Streptococcus in 3 of the 4 HCR subjects, and Lactobacillus in the 4th HCR subject. Genus Streptococcus and Lactobacillus were at lower levels or undetectable in the LCR subjects (see Fig. 6).

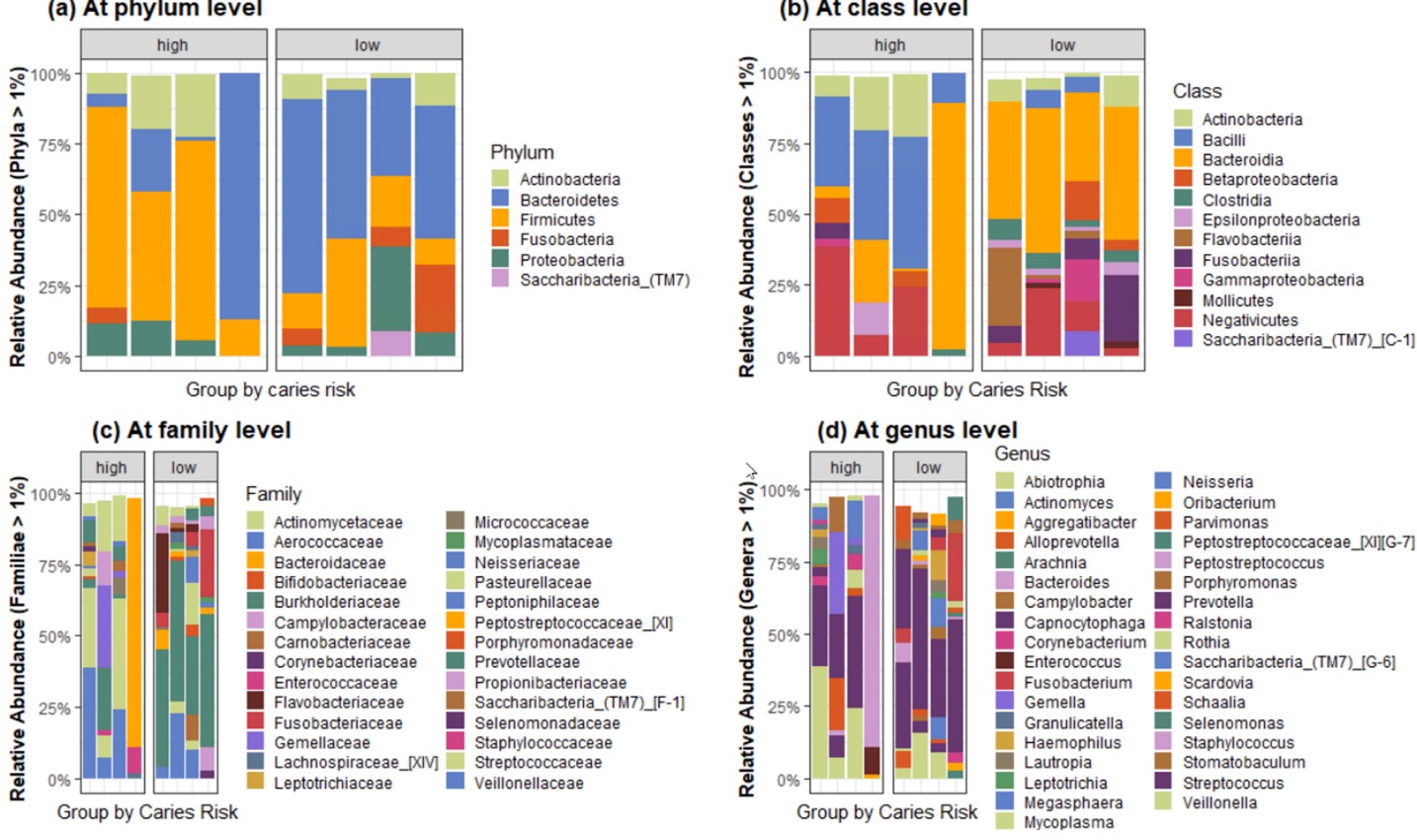

**Figure 3** Bar plots showing the composition of bacterial communities by caries risk at (A) phylum, (B) class, (C) family, and (D) genus level across all samples at >1% abundance.

**Table 5** Most abundant phylum, class, family, and genus.

| At phylum level | At class level | At the family level | At the genus level |
|---|---|---|---|
| Bacteroidetes (44.48%) | Bacteroidia (36.60%) | Prevotellaceae (27.88%) | Prevotella (24.83%) |
| Firmicutes (27.96%) | Negativicutes (12.16%) | Veillonellaceae (11.76%) | Veillonella (11.03%) |
| Proteobacteria (11.88%) | Bacilli (11.90%) | Flavobacteriaceae (8.44%) | Capnocytophaga (8.55%) |
| Actinobacteria (7.56%) | Flavobacteriia (8.25%) | Bacteroidaceae (8.23%) | Bacteroides (8.39%) |

## DISCUSSION

In this study, we show a significant association between BSI in pediatric HCT recipients and caries risk. Other studies report age (*Akinboyo et al., 2020*) and severity of mucositis (*Gudiol et al., 2014*; *Ustun et al., 2019*) as associated with BSI, but in our study groups, these associations no longer remained when dental caries risk was included in a multiple regression analysis. These findings indicate that the oral microbiome associated with caries risk is a significant risk factor for BSI in pediatric HCT patients.

The placement of dental restorations and extractions causes a transient bacteremia (*Bahrani-Mougeot et al., 2008*), so while it is possible that increased BSI was related to

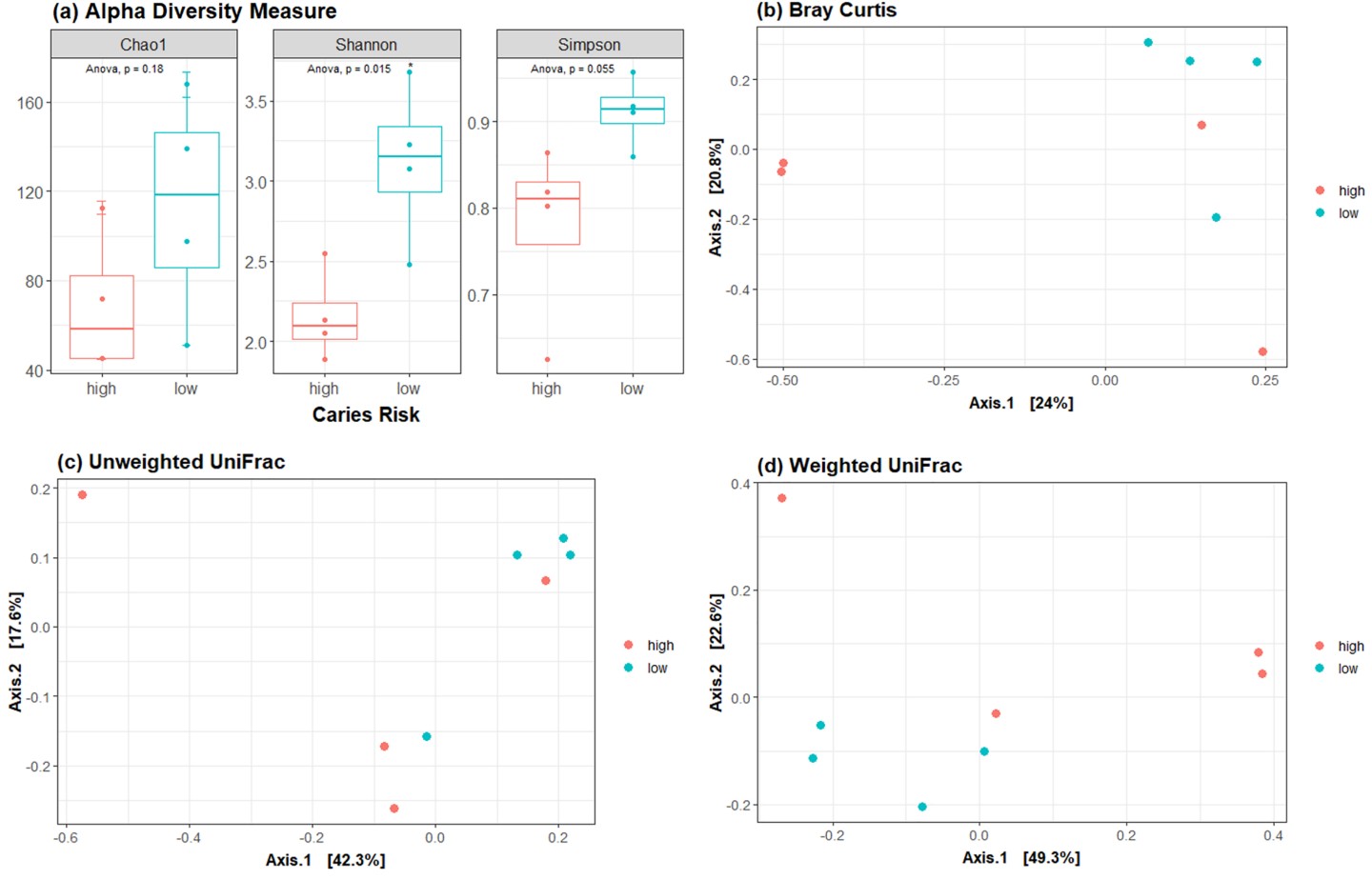

**Figure 4 Diversity metrics and principal coordinate analysis.** (A) Alpha diversity metric showing the difference in richness and evenness of communities. Beta diversity metrics showing the clustering by community composition by principal coordinate analysis using (B) Bray Curtis, (C) Unweighted UniFrac and (D) Weighted UniFrac methods.

bacteremia resulting from dental restorations prior to transplant, such transient bacteremia would have most likely resolved by the time of transplant. The placement of dental restorations does not reduce the cariogenic bacterial load in the mouth (*Philip, Suneja & Walsh, 2018*); that must be done through other approaches that target cariogenic bacteria.

We found no difference in beta diversity, but significantly decreased alpha diversity in the post-conditioning HCR subgroup, as compared to the LCR subgroup. Beta diversity compares microbial communities, whereas alpha diversity, also known as species diversity, compares the diversity of species within a community. While we did not compare microbial diversity pre conditioning, dental caries is not generally associated with changes in alpha diversity in the pediatric oral microbiome (*Jiang et al., 2016*). Conditioning regimens, which use chemotherapy and/or radiation can induce shifts in the oral microbiota (*Hong et al., 2019*; *Laheij et al., 2019*; *Vasconcelos et al., 2016*). Therefore, reduced alpha diversity in the HCR oral microbiome post conditioning suggests the possibility that the cariogenic microbiome has a unique ecological shift induced by
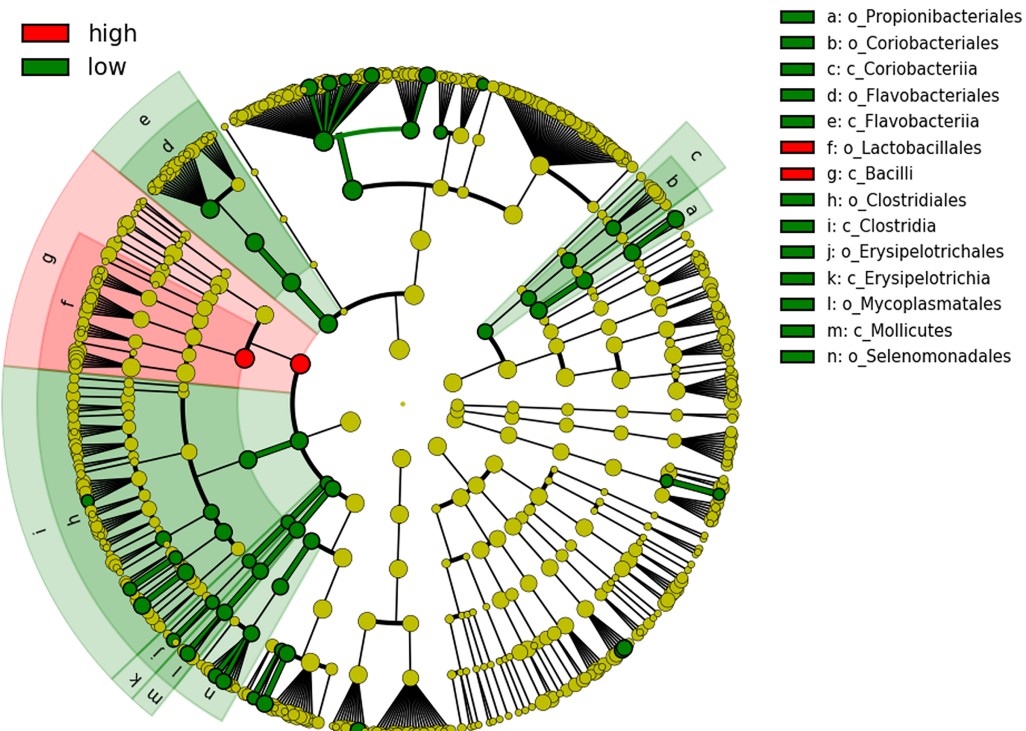

**Figure 5 Cladogram showing significantly different taxa at order and class level.** Caries risk status is delineated by the prevalence of oral bacteria. The six rings within the cladogram represent different levels within bacterial classification. Shaded circles refer to the significant taxa identified by LEfSe analysis. Red indicating bacterial groups enriched in the high caries risk group and green indicating those that are highly abundant in low caries risk group. Each circle's diameter is proportional to the taxon's abundance.

conditioning regimens. Lower oral microbiome diversity at preconditioning of autologous HCT patients has been found to be associated with a higher risk of relapse and worse survival (*de Molla et al., 2021*). Therefore, reduced oral microbiome diversity in HCR patients could also potentially impact long term transplant outcomes.

The HCR microbiome was significantly enriched in Lactobacillales. The order Lactobacillales contains viridans group streptococci, including *S. mutans* and *S. sobrinus*, and also lactobacilli, all of which are associated with dental caries (*Beighton, 2005*), but their mechanistic links to pediatric HCT morbidities remain unknown. The validity of our use of a retrospective assessment of pretransplant dental clearance records to assess caries risk, was supported by the finding that the relative abundance of bacteria from genus Streptococcus or Lactobacilli was enhanced in the HCR as compared to LCR subjects.

Xylitol, which was previously found to be associated with a reduced incidence of BSI in pediatric HCT recipients (*Badia et al., 2020*), targets the cariogenic bacteria, *S. mutans*. *S. mutans* does not act alone in the development of dental caries, but it alters the local environment by forming a favorable niche for other acidogenic and aciduric species to thrive (*Lemos et al., 2019*). Further studies using metagenomic sequencing will identify unique features of the cariogenic microbiome in HCT recipients; how it is altered by

**Table 6 Significantly different taxa in LCR as compared to HCR in both analysis (DESeq2, LEfSe).**

| log2FoldChange | Taxa |
| --- | --- |
| −4.813 | p_Firmicutes,c_Bacilli,o_Lactobacillales |
| −6.16817 | p_Firmicutes,c_Bacilli |
| 9.734868 | p_Bacteroidetes,c_Bacteroidia,o_Bacteroidales,f_Prevotellaceae,g_Prevotella,s_nigrescens |
| 11.36474 | p_Fusobacteria,c_Fusobacteriia,o_Fusobacteriales,f_Fusobacteriaceae,g_Fusobacterium,s_sp._HMT_203 |
| 8.819969 | p_Firmicutes,c_Negativicutes,o_Veillonellales,f_Veillonellaceae,g_Megasphaera,s_micronuciformis |
| 10.11979 | p_Bacteroidetes,c_Bacteroidia,o_Bacteroidales,f_Prevotellaceae,g_Prevotella,s_marshii |
| 10.46362 | p_Firmicutes,c_Mollicutes,o_Mycoplasmatales,f_Mycoplasmataceae,g_Mycoplasma |
| 8.713728 | p_Firmicutes,c_Negativicutes,o_Veillonellales,f_Veillonellaceae,g_Megasphaera |
| 9.631108 | p_Firmicutes,c_Clostridia,o_Clostridiales,f_Peptoniphilaceae,g_Parvimonas |
| 9.249647 | p_Firmicutes,c_Clostridia,o_Clostridiales,f_Peptostreptococcaceae_[XI] |
| 10.20735 | p_Firmicutes,c_Mollicutes,o_Mycoplasmatales,f_Mycoplasmataceae |
| 9.201216 | p_Firmicutes,c_Clostridia,o_Clostridiales,f_Peptoniphilaceae |
| 10.09428 | p_Firmicutes,c_Mollicutes,o_Mycoplasmatales |
| 4.250351 | p_Firmicutes,c_Erysipelotrichia,o_Erysipelotrichales |
| 4.855536 | p_Bacteroidetes,c_Flavobacteriia,o_Flavobacteriales |
| 8.66294 | p_Firmicutes,c_Mollicutes |
| 4.940307 | p_Bacteroidetes,c_Flavobacteriia |
| 3.530813 | p_Firmicutes,c_Erysipelotrichia |

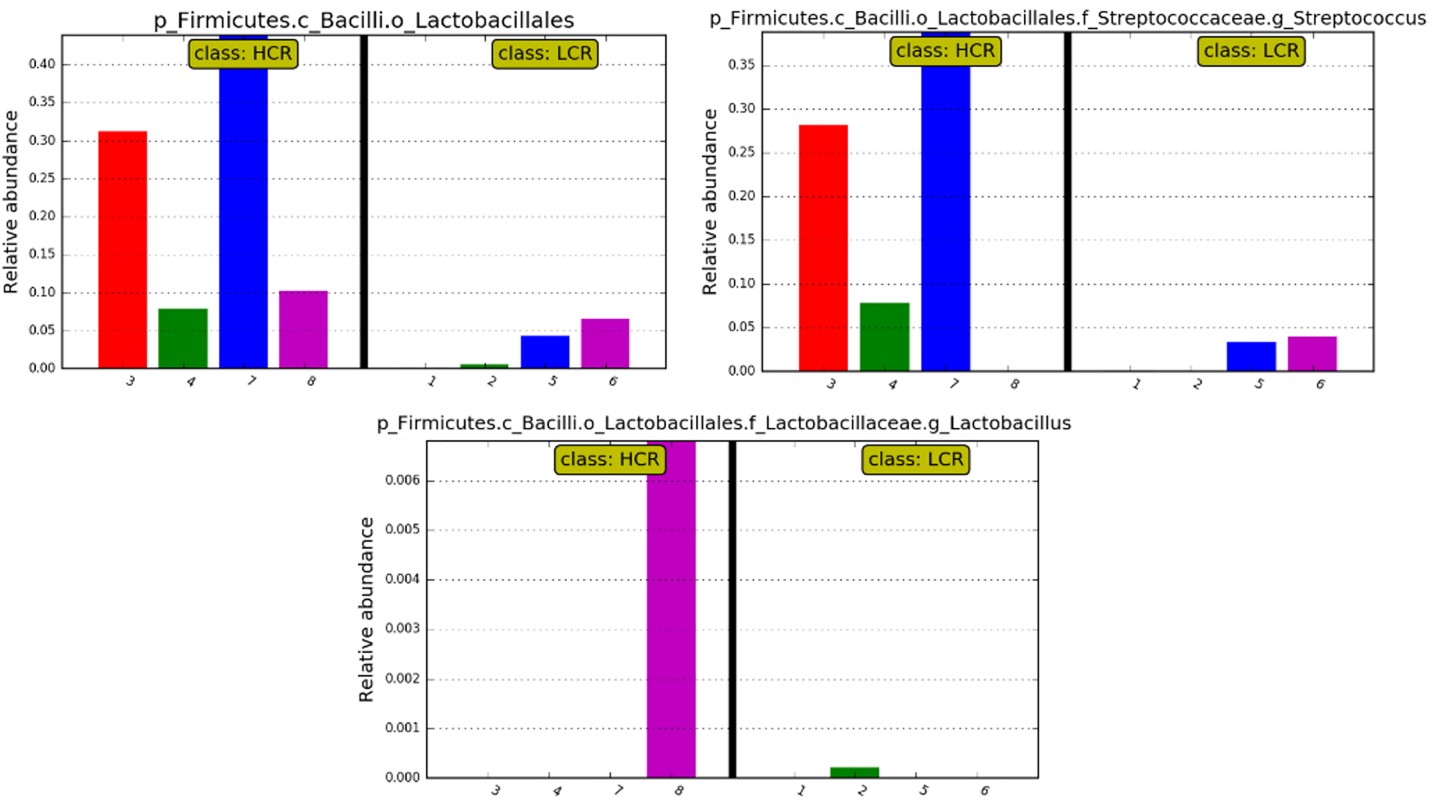

**Figure 6 Bar plots showing the differences of relative abundance.**

conditioning; and its association with other microbial niches, such as the oral mucosa where the BSI-associated *S. mitis* (*Doern & Burnham, 2010*) resides.

## CONCLUSIONS

Much is to yet to be understood about the oral microbiome of children with undergoing HCT; how changes in the oral microbiome, such as those that result in dental decay, affect the oral environment following conditioning and transplant; and how changes in the oral microbiome may affect BSI. The significant association between dental caries risk and BSI in our study population suggests an important new consideration in pediatric transplant patients.

### Funding
This funding was supported by the Center for Children Oral Health Research and T32DE007306. The funders had no role in study design, data collection and analysis, decision to publish, or preparation of the manuscript.

### Grant Disclosures
The following grant information was disclosed by the authors:
Center for Children Oral Health Research: T32DE007306.

### Competing Interests
The authors declare that they have no competing interests.

### Author Contributions
- Dawud abduweli uyghurturk conceived and designed the experiments, performed the experiments, analyzed the data, prepared figures and/or tables, authored or reviewed drafts of the article, and approved the final draft.
- Ying Lu performed the experiments, analyzed the data, prepared figures and/or tables, authored or reviewed drafts of the article, and approved the final draft.
- Janelle Urata performed the experiments, authored or reviewed drafts of the article, and approved the final draft.
- Christopher Dvorak analyzed the data, authored or reviewed drafts of the article, and approved the final draft.
- Pamela Den Besten conceived and designed the experiments, analyzed the data, authored or reviewed drafts of the article, and approved the final draft.

### Human Ethics
The following information was supplied relating to ethical approvals (*i.e.*, approving body and any reference numbers):
The University of California San Francisco granted Ethical approval to carry out the study within its facilities(15-18297).

## DNA Deposition

The following information was supplied regarding the deposition of DNA sequences:

The data is available at GEO: GSE199991.

## Data Availability

The data is available at GitHub:

– https://github.com/duyghurturk/MicrobiomeAnalysisWithR_BMT_PeerJ.
– https://github.com/duyghurturk/Qiime2Analysis_BMT_PeerJ.

## Supplemental Information

Supplemental information for this article can be found online at http://dx.doi.org/10.7717/peerj.14040#supplemental-information.

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
