# Peer review of "Dental caries as a risk factor for bacterial blood stream infection (BSI) in children undergoing hematopoietic cell transplantation (HCT)"

_PeerJ, doi:10.7717/peerj.14040_

## Round 0.1 · original submission · Major Revisions

Dear Authors

Kindly respond to the reviewer queries.

Best Regards
Dr Mallineni

·

Basic reporting

- The manuscript is well written and easy to understand. It is the perfect length for the subject material and is unambiguous. The literature references are sufficient. The figures and tables are easy to understand.

Experimental design

The aims of the study are within the scope of the journal.
The research question is well defined, relevant, and meaningful.
The methods are a bit unclear. Was this a retrospective analysis of the dental assessments? or was a dental examination done for the study?
Please include details on how mucositis was assessed and the timing of the assessments.

Validity of the findings

I would recommend a few additional data to be explained in the results:
- Please include the total number of patients approached for the study during the time.
- Please include additional information on the classification and dental assessments between the high-risk and low-risk cohorts. Perhaps a table or figure describing the dental findings/summary of the patients?
- Please include more detail on the timing of the infections. Additionally, the total number of infections patients incurred. Did the infections occur prior to engraftment?
- COuld you provide more detail on the patients that developed a BSI, in particular allogeneic vs. autologous transplant patients?

Additional comments

No other comments

Reviewer 2 ·

Basic reporting

A manuscript ‘Dental caries as a risk factor for bacterial blood stream infection (BSI) in children undergoing hematopoietic cell transplantation (HCT)’ contains original data obtained with a modern technique of the DNA metabarcoding. This manuscript reports on differences of oral microbiota in two groups of children with different dental caries risk underwent hematopoietic cell transplantation. However, in its current form the manuscript could not be accepted for publication in this journal by few reasons.
1) A hypothesis is not formulated at all. It should be formulated in the end of Introduction section.
2) Raw reads are unavailable, because they are not deposited in a open database. As an example, you can use SRA of NCBI for this.

Experimental design

In the present form the study is not reproducible. Methods should be described with sufficient information to be reproducible by another investigator.
1) Inclusion and exclusion criteria are not indicated for the clinical trial.
2) Characteristics of those 8 patients examined with DNA metabarcoding of oral microbiome should be given as the whole cohort (Table 1).
3) Clear criteria should be indicated for including patients from the whole cohort into the subgroups undergoing oral sampling.
4) Clear criteria for evaluation of a dental decay level should be described as well as its classification used.
5) Detailed description of a procedure for the oral sampling is required, including methods of DNA conservation, duration and temperature of transportation to the laboratory, etc.
6) Name of a DNA sequencer and description of DNA isolation, DNA library preparation, DNA sequencing with references and commercial names of reagent kits should be requested at the uBiome research sequencing laboratory and indicated in the manuscript. What kind of sequencing was used, pair-end or single-end? What was length of the raw reads?
7) If the V4 region of the 16S rRNA gene with size about 290 bp was sequenced using universal primers 515F and 806R with the pair-end 2*150 sequencing as described in Dionizio et al. 2021, why the authors denied reads merging into full-size contigs, and worked with short raw reads cutting them additionally up to 125 bp? Such procedure dramatically decreased possibility to determine the taxonomic assignment of reads especially at level of lower taxa of genera. Moreover, exact taxonomic identification of 125 bp OTUs at the genus level is unlikely in contrast to the statement of the authors that 98% of the OTUs were identified at the genus level (Line 162). Moreover, on the site of the HOMD databse it is indicated that among 774 oral bacterial species only 58% are officially named, whereas 16% unnamed but cultivated and 26% known only as uncultivated phylotypes. Thus, I ask the authors to explain their bioinformatics pipeline, to correct respective parts of the text, and to resolve these doubts.
8) Line 174 and below. Please exchange "taxonomies" by "taxa".

Validity of the findings

Conclusions are well stated, linked to original research question & limited to supporting results. But one inconsistency should be resolved. The main finding of thee study demonstrates that the HCR microbiomes are significantly enriched in representatives of order lactobacillales. But Figure 3 does not contain respective chart at the order level. Moreover, although the authors discussed role of S. mutans, S. sobrinus, and lactobacilli in dental caries development (Lines 210-217), they did not provide any data on differences of HCR and LCR oral microbiomes in terms of relative abundances of Streptococcus and Lactobacillus. Even there are no significant differences, this data should be analysed and discussed. I believe that filling the gap will improve validity of the findings.

---

## Round 0.2 · accepted · Accept

Dear Authors,

Congratulations. Thank you for considering PeerJ.

Best Regards

·

Basic reporting

The manuscript is clear and sustinct. I especially appreciate the length!

Experimental design

No concerns with the experimental design

Validity of the findings

The conclusions are sustained by the study results

Additional comments

no other concerns. The authors have addressed my previous comments.

Reviewer 2 ·

Basic reporting

A manuscript ‘Dental caries as a risk factor for bacterial blood stream infection (BSI) in children undergoing hematopoietic cell transplantation (HCT)’ contains original data obtained with a modern technique of the DNA metabarcoding. This manuscript reports on differences of oral microbiota in two groups of children with different dental caries risk underwent hematopoietic cell transplantation. The authors corrected all unclear fragments according to the reviewers' recommendations. Thus, the manuscript in its current form could be accepted for publication in this journal.
I have only one minor recommendation to the authors. According to current rules on publishing of biomedical data based on protein or nucleic acids sequences, your raw reads should be available. For this you should to deposit them in a open database such as SRA of NCBI (https://submit.ncbi.nlm.nih.gov/subs/sra/) or ENA of EMBL-EBI (https://www.ebi.ac.uk/ena/browser/home) or somewhere else.

Experimental design

The hypothesis is formulated. The methods used meet the aim of the study. In the present form the study is reproducible. Methods are described with sufficient information to be reproducible.

Validity of the findings

Impact and novelty are assessed. Meaningful replication encouraged where rationale & benefit to literature is clearly stated. All underlying data have been provided; they are robust, statistically sound, & controlled. Conclusions are well stated, linked to original research question & limited to supporting results.